# Intercellular communication between artificial cells by allosteric amplification of a molecular signal

Bastiaan C. Buddingh'[1,2], Janneke Elzinga [3] & Jan C. M. van Hest [1,2,4 ✉]

Multicellular organisms rely on intercellular communication to coordinate the behaviour of individual cells, which enables their differentiation and hierarchical organization. Various cell mimics have been developed to establish fundamental engineering principles for the construction of artificial cells displaying cell-like organization, behaviour and complexity. However, collective phenomena, although of great importance for a better understanding of life-like behaviour, are underexplored. Here, we construct collectives of giant vesicles that can communicate with each other through diffusing chemical signals that are recognized and processed by synthetic enzymatic cascades. Similar to biological cells, the Receiver vesicles can transduce a weak signal originating from Sender vesicles into a strong response by virtue of a signal amplification step, which facilitates the propagation of signals over long distances within the artificial cell consortia. This design advances the development of interconnected artificial cells that can exchange metabolic and positional information to coordinate their higher-order organization.

[1] Department of Chemical Engineering and Chemistry, Eindhoven University of Technology, PO Box 513, 5600 MB Eindhoven, the Netherlands. [2] Institute for Complex Molecular Systems, Eindhoven University of Technology, PO Box 513, 5600 MB Eindhoven, the Netherlands. [3] Radboud University, PO Box 9102, 6500 HC Nijmegen, the Netherlands. [4] Department of Biomedical Engineering, Eindhoven University of Technology, PO Box 513, 5600 MB Eindhoven, the Netherlands. ✉email: j.c.m.v.hest@tue.nl

Living systems generally do not operate in isolation, rather they are often intimately connected as cooperators or competitors[1,2]. Intercellular communication is critical to coordinate the behaviour of individual cells in multicellular communities, whether in multicellular organisms that use signalling to organize, synchronize and differentiate their specialized tissues[3], or in consortia of prokaryotes that control their behaviour based on the total population density[4]. Biological cells utilize a variety of signalling processes to exchange information with each other and to sense their environment. Signals can be electrical or mechanical in nature, yet most are based on diffusible chemical messengers which are secreted from the senders and recognized by the receivers through autocrine, paracrine or endocrine paths[5,6].

In order to understand how cells coordinate their behaviours through intercellular communication, the engineering of existing or entirely new synthetic communication networks has been undertaken within the field of synthetic biology. In a top-down approach, existing cell lines have been engineered to express synthetic communication pathways that are orthogonal to the cell's natural signalling networks, which enabled non-natural signalling and regulatory pathways in biological cells[7–10]. Although functionally complex, these synthetic information processing networks rely heavily on the intrinsic biological machinery and can, therefore, be regarded as additional layers of signal processing, rather than a fully engineered system. Bottom-up approaches have used synthetic cell-like compartments such as liposomes, polymersomes, coacervates and colloidosomes as the chassis for artificial cells[11,12]. Advances in this field have now led to the creation of artificial cells with increasing complexity—integrating information processing networks[13], metabolic functions[14] and functional behaviour like membrane growth[15] and molecular sensing[16]. Until very recently, however, the focus of artificial cell engineering has been on the design of discrete systems that function in isolation.

In biological multicellular communities, on the other hand, the interaction between different constituents is critical for the coordination of spatially separated functions that support the emergence of higher-order structures[17,18]. In the development of artificial cells, however, the construction of such communities that can communicate and self-organize into synthetic tissues has remained underexplored to date. The potential of artificial cells to develop new technologies based on biochemical information processing was explicitly put forward several years ago[19], and since then several hybrid communication systems between artificial cells and biological cells have been developed. These typically relied on passive diffusion of the signalling molecule over the synthetic membrane, and harnessed the biological cell's resources to transduce a signal into the expression of a reporter gene[20–24]. Recently, the first communities fully comprised of artificial cells were reported that can communicate by diffusible chemical signals to coordinate their behaviour. Elegant examples of signal processing were demonstrated using communities of synthetic particles with information-encoding DNA sequences[25] or supramolecular structures[26,27] grafted onto the particles, in which signal processing was performed in solution rather than within the compartments. Contrarily, information exchange over the membranes of synthetic compartments was more restricted in the molecular nature of the messenger and, therefore, relied on small molecules such as hydrogen peroxide[28–31] or the expression of a pore-forming protein[32,33]. Juxtacrine-like signalling has been achieved using droplet interface bilayers[34,35], which enabled the controlled geometrical patterning of sender and receiver compartments and afforded precisely modulated spatiotemporal signal gradients over several connected compartments[36,37]. Paracrine-like signalling between individual artificial cells by freely diffusing chemical messengers, on the other hand, has been demonstrated using molecular messengers like DNA[13] and mRNA or proteins[38]. Such large signalling molecules, however, required porous compartments to allow the exchange of molecular information, yet also selective sequestration of messengers and machinery inside the compartments to permit differentiation of subpopulations of compartments. Although the collective behaviour that can be generated is becoming increasingly complex with signalling circuits that can now program feedback loops[13,36], amplification and logic tasks[13] and differentiation[36,38,39], most systems rely on cell-free protein expression[36,38] or DNA-strand displacement circuits[13] rather than chemical signals based on small molecules and protein-based signal processing that is more reminiscent of natural signalling pathways. Furthermore, it is crucial that the signal transduction includes a strong amplification step to relay low concentrations of secreted chemical signals into an effective intracellular process. Natural signalling pathways, like endocrine signalling by hormones, rely on signal amplification to translate the stochastic stimulation elicited by dilute molecular messengers into a strong, amplified intracellular response[5,40].

Here, we present a communication pathway between two populations of artificial cells that demonstrates the importance of signal amplification for efficient signalling by secreted chemicals. This is the first allosterically activated communication network in artificial cells, and fully relies on protein machinery and small molecules as chemical information agents. We study the interaction between artificial cells that produce a chemical signal (Senders) and another population that is programmed to perceive the chemical signal, transduce it, and produce an internal response (Receivers). This allosterically activated platform is generic and, because of its enzymatic basis, allows facile extension to other biological functions.

## Results

**An allosterically activatable communication pathway.** In order to implement an intercellular communication platform, we designed two populations of artificial cells. First, Senders respond to an external trigger and process this into a signalling molecule that is released. This signalling molecule diffuses through the extracellular environment until it reaches a Receiver, which recognizes the chemical signal and activates in response to the information relayed by the Sender. Upon secretion from the Sender, however, the chemical signal is highly diluted; depending on the distance between the Senders and Receivers, the concentration of signalling molecules may be extremely low upon reaching a Receiver. Signal amplification is therefore highly important to achieve adequate activation of the Receivers.

We opted to use giant unilamellar vesicles (GUVs) composed of phospholipids to create the communities of artificial cells, because of their semipermeable membrane and their compatibility with biological parts, which allows the loading of virtually any (macro)molecule with high efficiency[41]. To build a communication network capable of signal amplification, we turned to allosteric activation, which allows the modulation of the activity of certain enzymes by small molecular activators. We selected glycogen phosphorylase b (GPb), an enzyme that plays a key role in glycogen homeostasis and can switch between a tense conformational state that has low activity and a relaxed state with high turnover of glycogen. The switch to the high-activity state is induced by the binding of adenosine 5′-monophosphate (AMP), which is an important metabolic intermediate that also regulates various other signalling pathways[42,43]. By loading the Receivers with GPb, we endowed them with an activation switch that processes low concentrations of AMP into an amplified

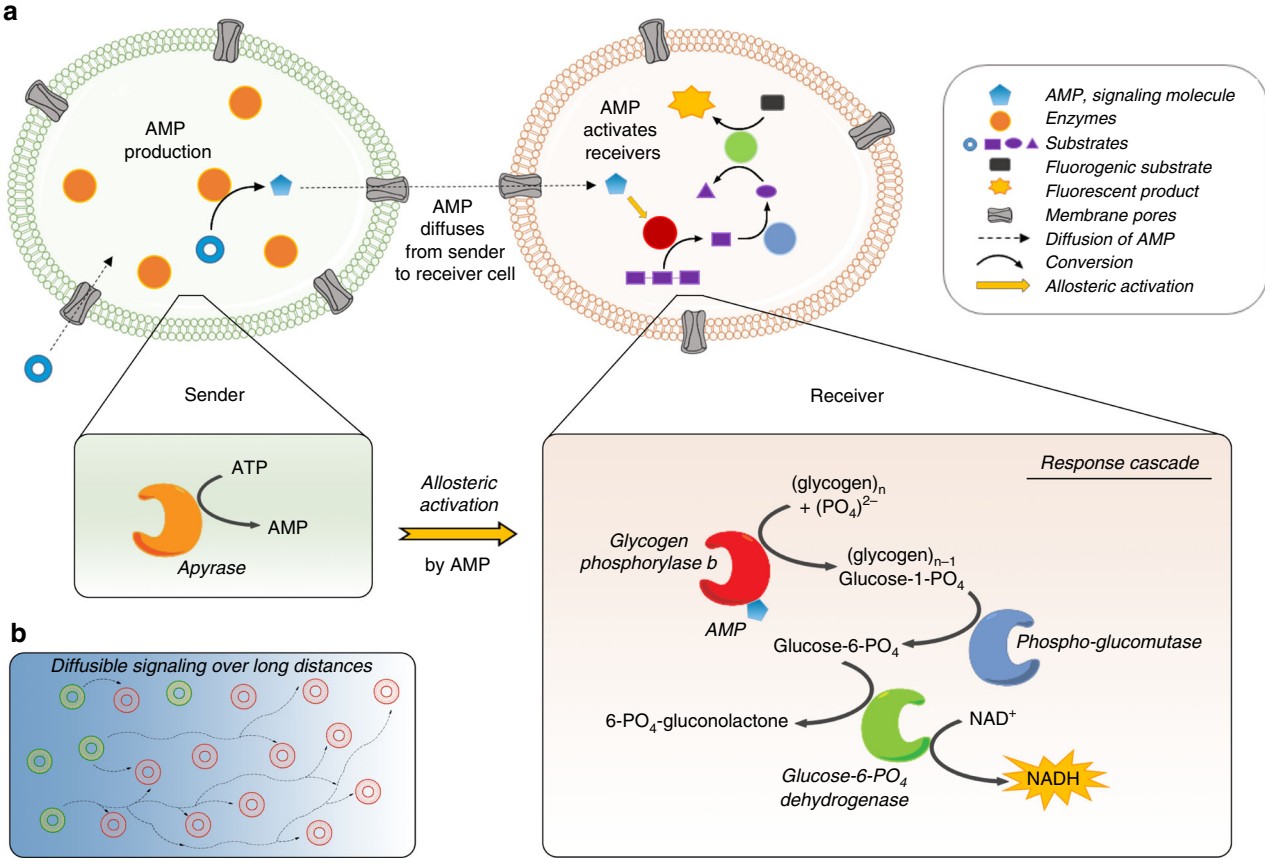

**Fig. 1 Design of a communication pathway capable of signal amplification between two populations of artificial cells. a** A signalling molecule (AMP) is produced by the Sender population and diffuses to the Receiver population, where it is perceived, processed, and an internal response is generated. The Response Cascade encapsulated in the Receivers is capable of signal amplification by allosteric activation of the first enzyme in a cascade that ultimately generates NADH, which is an important anabolic metabolite and here specifically used as fluorescent reporter of successful communication. **b** The signal amplification in the Receivers permits communication over long distances, despite strong dilution of the chemical signal in the external milieu.

output. Adding two downstream enzymes from the pentose phosphate pathway (phosphoglucomutase; PGM, and glucose-6-phosphate dehydrogenase; G6PDH), reconstitutes a metabolic pathway that ultimately produces NADH, an important metabolite in the energy balance of biological cells (Fig. 1a).

The allosteric activation of GPb by AMP is of key importance, as it provides a powerful amplification step that transduces low concentrations of AMP into a strong activation of the Receivers, and hence a high NADH output. To generate AMP in the Senders, GUVs were loaded with the enzyme apyrase, which readily generates AMP from ADP or ATP by the sequential hydrolysis of one or two phosphate groups, respectively. However, AMP cannot readily cross the phospholipid bilayer of GUVs because of its charged phosphate group. To facilitate the exchange of AMP between populations of GUVs, we used a membrane pore protein, α-haemolysin (αHL), to allow the passage of molecules <2 kDa[44].

Thus, Sender cells are designed to convert an external trigger (ATP or ADP) into a signalling molecule (AMP) that can diffuse to a population of Receiver artificial cells, which even at low concentrations of signal still generate an amplified output upon signal recognition (Fig. 1a, b).

**Efficient signal amplification**. To efficiently activate the Receivers and achieve high signal amplification, the three-enzyme cascade described above (Response Cascade) was first tested and optimized in bulk reactions. Parameters like enzyme concentrations, buffer composition and pH were varied to achieve high

sensitivity to AMP, sufficient NADH production, and suitable reaction times (Supplementary Note 1 and Supplementary Figs. 1–5). The Response Cascade was optimized such that the GPb concentration was the critical parameter for tuning the activation kinetics, and the availability of the final substrate (NAD$^+$) controlled the amplitude of the output at constant input strength (Fig. 2a–c).

The Response Cascade has three inputs that can be used to activate it: AMP, phosphate and glycogen. Whereas AMP is an allosteric activator of GPb and transforms the enzyme into an active state, phosphate and glycogen are both substrates of GPb and are fully converted and further processed by downstream enzymes. Due to its allosteric effect, AMP proved to be a strong activator of the Response Cascade with up to 80-fold increase of NADH production compared to the background activity in absence of AMP (Fig. 2d). This strong activation was translated into high signal amplification, producing up to 10 equivalents of NADH (output) compared to the AMP used to activate the module (input) (Fig. 2e). The pronounced effect of signal amplification was also effectively illustrated by comparing the activation of the Response Cascade by AMP (allosteric activation) or by phosphate (substrate activation) under equivalent conditions. Allosteric activation by varying levels of AMP generated a faster and stronger NADH output than substrate activation by corresponding phosphate levels (Fig. 2f, g). While the phosphate is converted until it is depleted, the AMP is not processed but remains bound to the activated GPb, ensuring the continuous conversion of substrate.

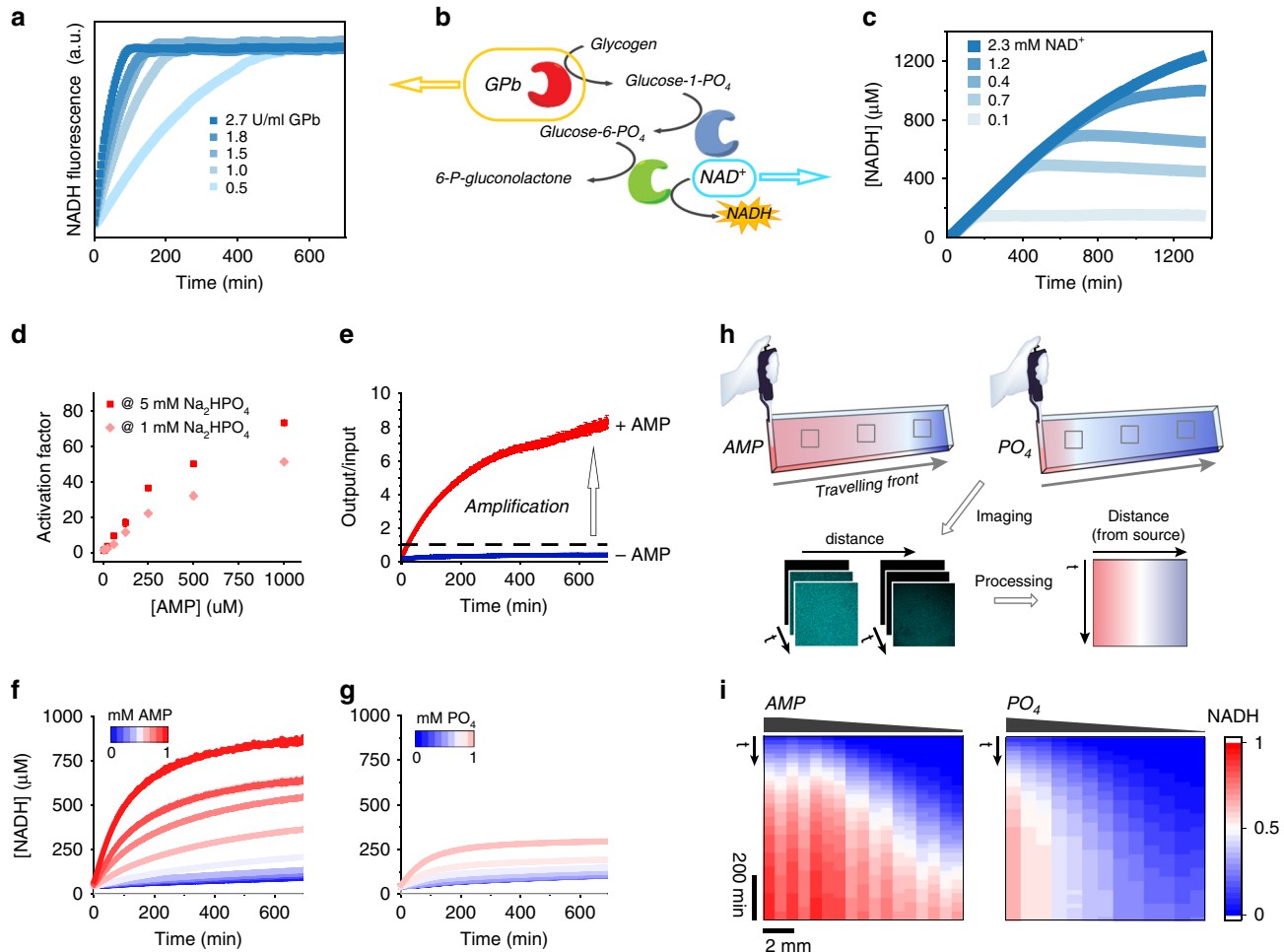

**Fig. 2 Amplification of the AMP signal by activation of a three-enzyme Response Cascade. a–c** Tuneable amplification of the AMP signal by the Response Cascade (**b**): (**a**) the GPb activity (0.5–2.7 U ml⁻¹) dictates the kinetics of the output generation; (**c**) the concentration of NAD⁺ (0.1–2.3 mM) regulates the amplitude of the NADH output. **d,e** AMP allosterically activates GPb, which increases the speed of NADH production compared to the inactive Response Cascade (expressed as the activation factor) (**d**), and results in signal amplification, as evidenced by a high ratio of NADH output per AMP input (60 μM) (**e**). **f,g** The importance of signal amplification is clear when allosteric activation by AMP (**f**, in the presence of 1.0 mM $Na_2HPO_4$) is compared to substrate activation by $Na_2HPO_4$ (**g**, in the presence of 1.0 mM AMP). AMP activation is much stronger than when equivalent levels of $Na_2HPO_4$ are employed. **h,i** Spatiotemporal activation of the Response Cascade in a channel device by AMP or $Na_2HPO_4$ diffusing from a point source. **h** The production of NADH was followed by confocal laser scanning microscopy (CLSM) in segments of the channels as a function of time. **i** Kymograms constructed from the CLSM micrographs plot the normalized NADH concentration along the length of the channel as a function of time. 1 mM AMP activates more and over longer distances when added as a point source to a channel loaded with the Response Cascade containing 1 mM $Na_2HPO_4$ (left) than 1 mM $Na_2HPO_4$ added as a point source to a channel containing the Response Cascade with 1 mM AMP (right).

We next investigated the effect of signal amplification on spatiotemporal activation patterns. We first studied the activation of the Response Cascade homogeneously loaded in a channel device, where either AMP or $Na_2HPO_4$ was added as a point source to one side of the channel (Fig. 2h). Figure 2i shows that AMP activated the Response Cascade throughout the channel in a distance-dependent manner. When phosphate was added as a point source, however, the activation was attenuated at long distances because the phosphate is converted along its diffusion path and its concentration decreases sharply over a short distance.

**Signal amplification in Receiver GUVs**. After establishing the desired allosteric response in bulk conditions, the next step was to compartmentalize the Response Cascade. To generate giant liposomes loaded with the Response Cascade the droplet transfer technique was employed, because it affords efficient

encapsulation of complex cargoes and typically produces thousands of GUVs[41]. Generally, a sucrose/glucose gradient is used to make the GUVs sediment, facilitating their observation by microscopy; because of the inhibitory effect of glucose on the Response Cascade, however, here it was replaced by sorbitol without affecting the GUV production. Furthermore, α-haemolysin was added to the observation chamber containing the GUVs, where it spontaneously self-assembled into membrane pores that facilitate the entrance of AMP by diffusion. To compensate for the efflux of crucial substrates from the Receivers, the solution containing the Receivers was supplemented with the necessary substrates. We employed time-lapse confocal laser scanning microscopy (CLSM) to follow the activation of the Receivers by recording the production of the fluorescent output marker NADH. When 1.0 mM AMP was added to these Receivers, the Response Cascade activated rapidly, as evidenced by their strong NADH fluorescence (Fig. 3a). As can be seen from

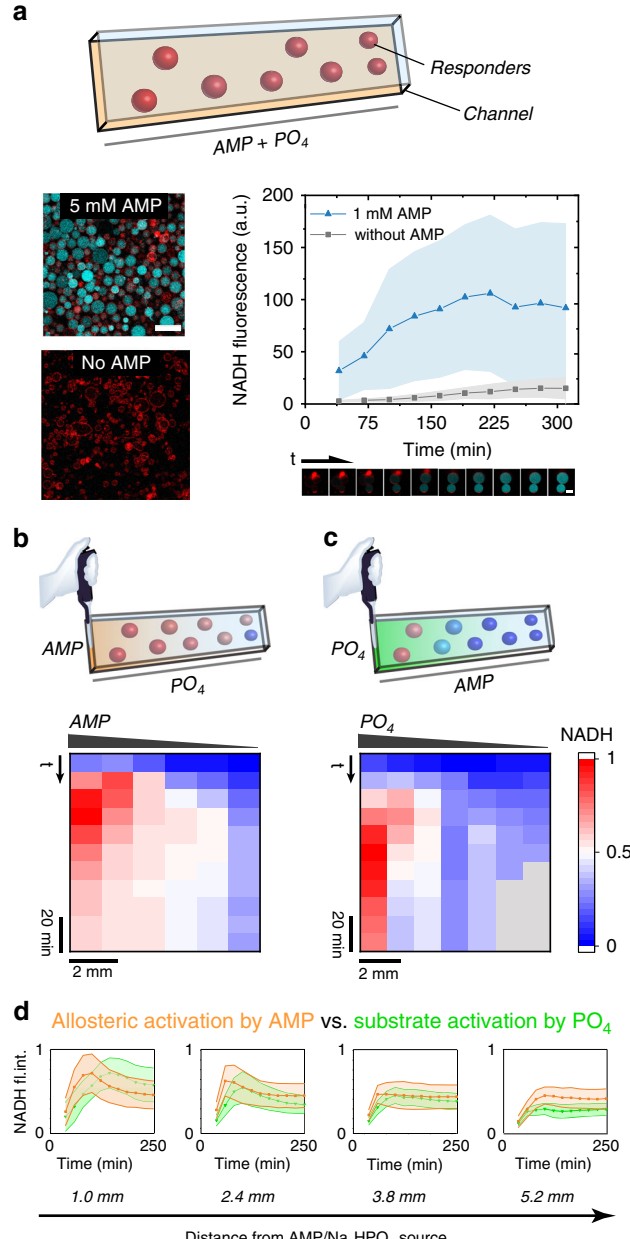

**Fig. 3 Activation of Receiver GUVs and the effect of signal amplification.**
**a** Receiver GUVs were incubated with solutions of either AMP and $Na_2HPO_4$ or $Na_2HPO_4$ alone. The activation of the Receivers was monitored by time-lapse CLSM, which tracked the production of fluorescent NADH (cyan). Receiver GUVs were labelled with DOPE-(Lissamine Rhodamine B) (red). When Receiver GUVs were incubated with AMP their Response Cascade was activated, whereas in absence of AMP the Receivers did not activate. Scale bars represent 50 µm. Error bars depict the SD from 23 GUVs, tracked in the CLSM micrographs. **b**, **c** Kymograms of the activation of Receivers by AMP or $Na_2HPO_4$ diffusing from a point source, with schemes showing the experimental set-up. 1 mM AMP activated over longer distances when added to the Receivers in presence of 1 mM $Na_2HPO_4$ (**b**) than 1 mM $Na_2HPO_4$ added to the Receivers in presence of 1 mM AMP (**c**). For each position the NADH production of >40 Receivers was analyzed and normalized. **d** Side-by-side comparison of the production of NADH in Receivers with either AMP or $Na_2HPO_4$ added as a point source (orange and green, respectively). Each panel displays the Receivers' activation at equivalent distances from the AMP or $Na_2HPO_4$ point sources (1.0 mm, 2.4 mm, 3.8 mm, 5.2 mm from point of AMP/$Na_2HPO_4$ addition). At every distance, AMP activated the Receivers more than $Na_2HPO_4$. The decrease in NADH fluorescence after the initial activation was caused by gradual efflux of NADH through the αHL pores. Error bars depict the SD.

**Senders activate Receivers in artificial cell communities**. The experiments in Figs. 2–3 demonstrate that the Response Cascade is capable of strong signal amplification and the Receiver GUVs are activated by AMP. We next investigated the transduction of a signal from one population of artificial cells (Senders) to another (Receivers). Both populations were mixed into a community with the Senders loaded with apyrase and the Receivers with the Response Cascade. To ensure that AMP production occurs within the Senders only, the GUVs were carefully washed after formation until all unencapsulated apyrase had been removed from the outer solution (Supplementary Fig. 9). To trigger signalling, ATP was added, which was converted by the Senders into AMP and released from the GUVs. In the presence of ATP, activation of the Receivers was observed, with a delay to allow the conversion of ATP into AMP by the Senders (Fig. 4a–c). Indeed, when ADP was supplied to the community the delay was reduced (Fig. 4d, e), which is attributed to the single hydrolysis of phosphate groups to convert ADP into AMP versus the sequential hydrolysis of two phosphates for ATP. Negative controls without either ATP, ADP or apyrase did not activate, which confirmed that the activation of the Receivers is dependent on the AMP signal generated by the Senders (Fig. 4f, g and Supplementary Figs. 10–11).

**Spatially propagating signalling fronts**. As described above, the strong signal amplification engineered into this pathway permits the activation of Receivers over large distances without the strong dampening observed for substrate activation (Fig. 3b–d). To demonstrate the efficient signalling of a small group of Senders over large distances, we loaded Receivers throughout a channel device that contained a cluster of Senders on one end. Upon triggering the Senders with ATP, a gradient of AMP was generated, which spread along the Receiver population and activated them in a distance-dependent manner (Fig. 5a–c). Importantly, the amplification of the AMP signal by the Response Cascade allows for efficient signalling over distances as large as 200 times the diameter of the Senders.

**Selective signalling to receptive artificial cells**. In multicellular communities, selective transduction of signals is of high importance for the regulation of collective behaviour. As such, different

the analysis of multiple GUVs, there was some heterogeneity in the NADH fluorescence amongst the Receivers. This is attributed to small variations in GUV loading as well as different degrees of αHL insertion, which resulted in variable kinetics for the influx of AMP and efflux of NADH (Supplementary Notes 2–3 and Supplementary Figs. 6, 12–14). In the absence of AMP, however, the Receivers clearly did not activate.

To study the activation of the Receivers by an AMP point source—reminiscent of the local production of signalling molecule by a small cluster of Sender GUVs—the channel device was loaded with Receivers and AMP was added to one side of the channel (Fig. 3b–d). The NADH production in the Receivers was recorded and extracted from the CLSM micrographs and plotted as a kymogram (Supplementary Figs. 7, 8). Compartmentalized into the GUVs, the Response Cascade displayed similar behaviour as in the bulk reactions in Fig. 2h, i. AMP could activate the Receivers over larger distances than phosphate, clearly revealing the importance of signal amplification for diffusive chemical signalling to artificial cells.

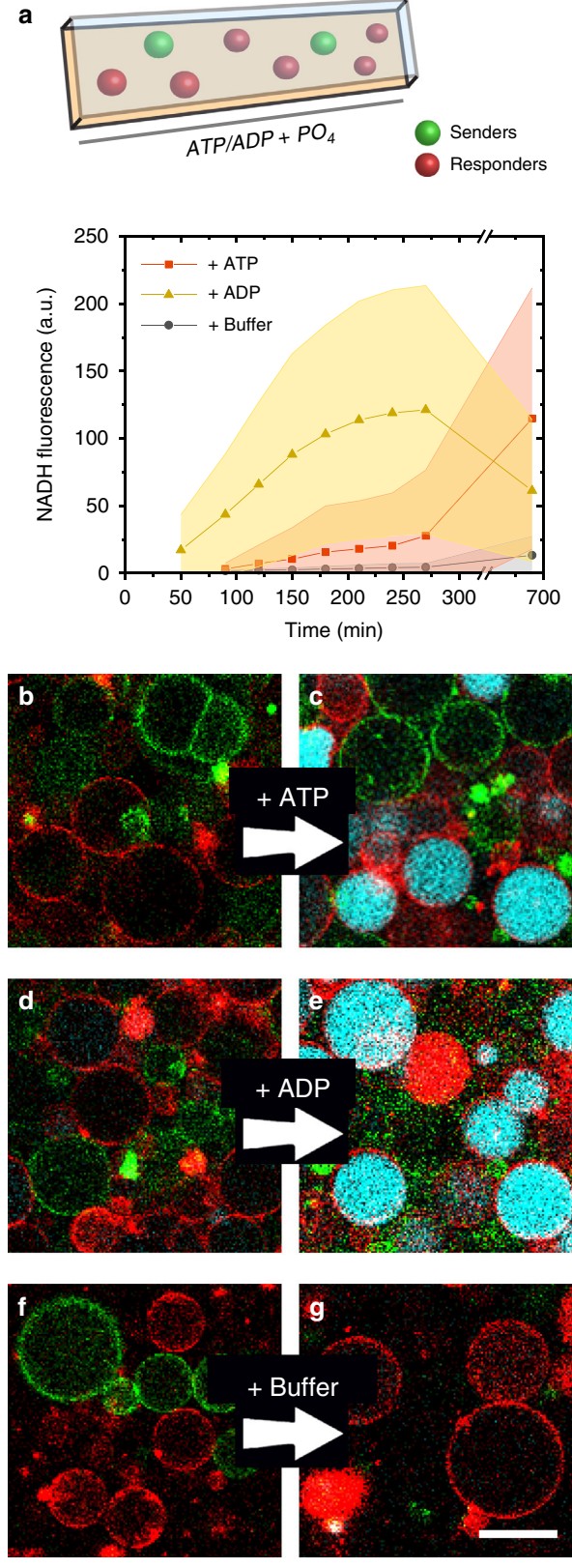

**Fig. 4 AMP-mediated signalling from Senders to Receivers. a** Senders and Receivers were mixed and incubated with ATP, ADP or buffer, as well as $Na_2HPO_4$. Both ATP and ADP (2.6 mM) were converted by Senders into AMP, which in turn activated the Receivers. As ATP was converted into AMP via ADP, the delay for activation of the Receivers was longer than when ADP was added directly. When ADP and ATP were both omitted, the Senders did not activate the Receivers. The decrease in NADH fluorescence after the initial activation was caused by gradual efflux of NADH through the αHL pores. Error bars depict the SD from 25 Receivers. **b–g** CLSM micrographs corresponding to the graph in (**a**). Receiver GUVs were labelled with DOPE-(Lissamine Rhodamine B) and Sender GUVs with DOPE-carboxyfluorescein. Overlay of NADH fluorescence (cyan), Sender GUVs (green) and Receiver GUVs (red). **b, d, f** $t = 50$ min after ATP/ADP/buffer addition; **c** $t = 690$ min; **e, g** $t = 270$ min. Scale bar represents 30 μm.

When ATP was supplemented to the Senders, the AMP they produced diffused into the extracellular solution and reached the two subtypes of Receivers. The receptive Receivers were activated by the increased levels of AMP and produced a strong NADH output, whereas the unsusceptible Receivers did not activate at all (Fig. 6b, d). This demonstrates that the signal produced by the Senders can selectively activate a receptive subpopulation of Receivers. To prove that the selective activation is indeed AMP-dependent, we also followed the activation of both subtypes of Receivers in the absence of ATP, such that no AMP was produced. Due to the low background activity of constitutively active GPa (Supplementary Note 1), a weak NADH signal could be observed, but activation of the Response Cascade was not observed for either subtype (Fig. 6c).

## Discussion

Although communication by diffusing chemical signals is widely used in nature to regulate multicellular behaviour, the engineering of synthetic, analogous systems remains challenging and is little explored[45,46]. The handful of designs that explore communication between artificial cells mostly rely on signalling molecules that (i) can induce protein expression in cell lysates[32,36,38] (ii) are small, uncharged substrates in model enzymatic cascades (most notably hydrogen peroxide)[28–31], (iii) or use DNA oligonucleotides or entire proteins to program a highly selective signal[13,25,38]. While nucleic acids and proteins do contribute to intercellular signalling, small molecules constitute the main class of information carriers secreted by cells, and they are recognized by proteins (receptors) that induce an internal enzymatic or genetic response. The communication system engineered here operates along these lines, as it uses enzymes and metabolites to signal between populations of artificial cells. Notably, this would facilitate its integration with other enzymatic pathways to extend the Response Cascade with functional behaviour like growth and differentiation.

One potential drawback of these enzymatic pathways, however, is their currently limited programmability compared to DNA-based circuits[13,25,47]. Modelling of the enzymatic reactions used here by differential equations would enable the simulation of signal transduction in space and time to predict the relevant parameter space for highly efficient signal propagation and feedback mechanisms. Although enzymatic networks have been successfully modelled in open, homogeneous systems[48,49], compartmentalization complicates their correct simulation due to extrinsic stochastic effects—such as differential solute-partitioning between individual vesicles[50], the existence of subpopulations of compartments with in- or decreased permeability[51,52], and complex interactions between components of the network and the compartment. Specifically, the heterogeneity in the activation kinetics of the Receivers in this study, attributed to both the diversity in loading of the GUVs and the limited control over

cell types have distinct susceptibilities to chemical signals to allow the selective activation of certain subpopulations. To mimic this behaviour in a community of artificial cells, we mixed Senders with two subpopulations of Receivers that contained either the full Response Cascade or the Cascade without GPb, and were, therefore, either receptive or unsusceptible to AMP (Fig. 6a).

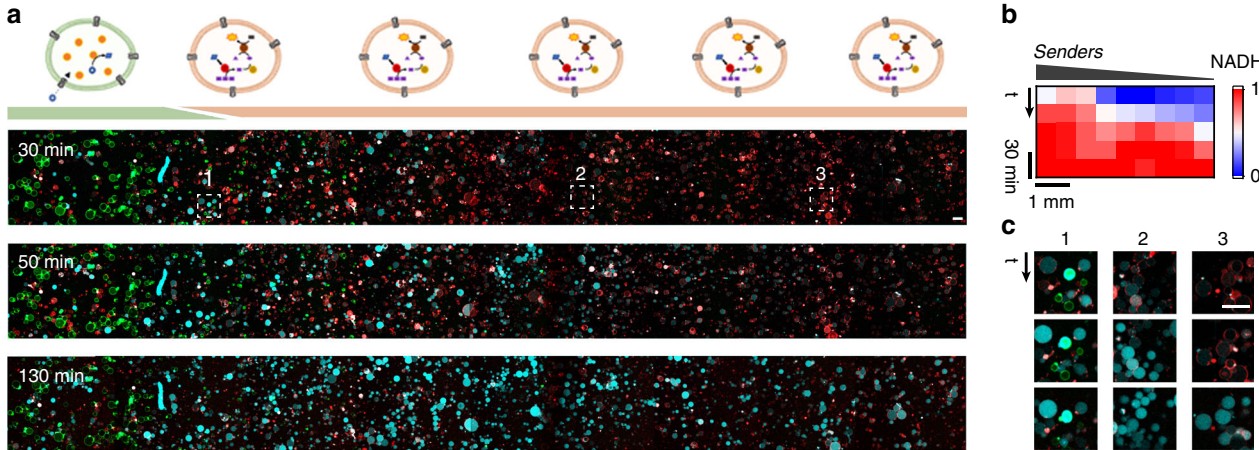

**Fig. 5 Spatially propagating signalling fronts in communities of Senders and Receivers. a** The Senders form a colony that was triggered by ATP to send an activation signal to the population of Receivers. From the cluster of Senders, a signalling front spread throughout the 5 mm long channel and activated the Receiver GUVs in a distance-dependent fashion. Overlay of NADH (cyan), Sender GUVs (green), Receiver GUVs (red). Scale bars represent 50 μm. **b** Kymogram of the activation of the Receivers by the signalling front, as measured by the relative NADH fluorescence throughout the channel at various time points. **c** Individual GUVs from the positions marked in (**a**), which demonstrate the distance-dependent activation of the Receivers.

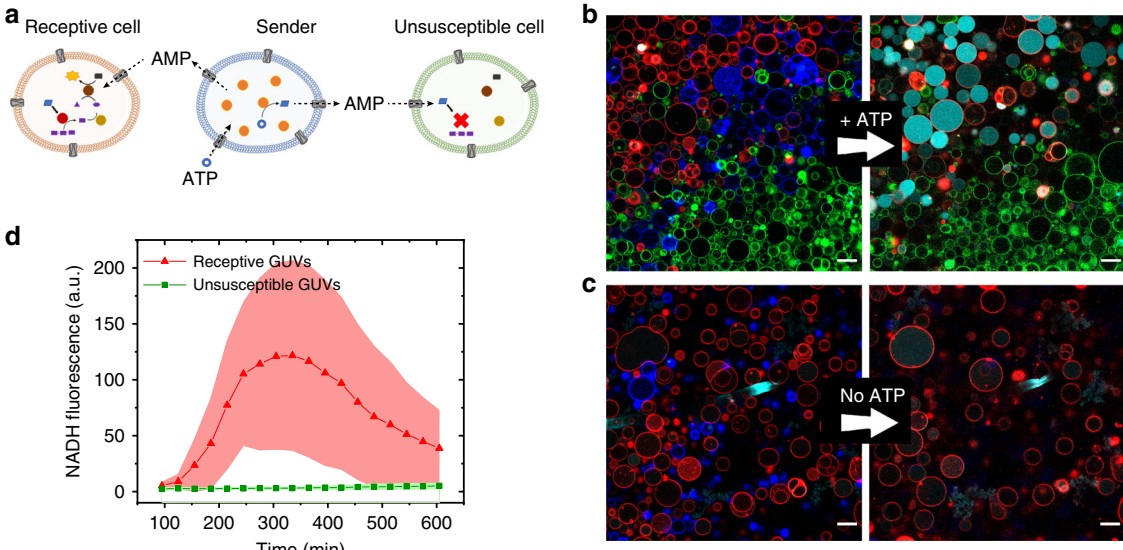

**Fig. 6 Selective signalling to receptive artificial cells. a** Differential activation of receptive and unsusceptible Receivers by ATP-triggered Senders. Only the receptive Receivers can be activated by the Senders, whereas the unsusceptible Receivers cannot recognize the signal and remain inactive. **b, c** CLSM micrographs of Senders and Receivers triggered with 2.6 mM ATP (**b**), or without ATP (**c**); before ATP and $Na_2HPO_4$ addition (left) or 270 min after (right). Overlay of NADH (cyan), Sender GUVs (blue), receptive Receivers (red), unsusceptible Receivers (green). Scale bars represent 30 μm. After 270 min the blue Cy5-signal had decreased due to bleaching, but the Senders were still present. **d** Differential activation of the two subtypes of Receivers, extracted from the time-lapse CLSM micrographs. Error bars depict the SD from 15 Receivers. The decrease in NADH fluorescence after the initial activation was caused by gradual efflux of NADH through the αHL pores.

permeability offered by a generic membrane pore like α-haemolysin, impeded their mathematical description[50,52]. Microfluidic techniques are envisioned to alleviate this issue, as they can produce more homogeneous populations of GUVs. This would simplify modelling of the current enzymatic circuits to predict the emergence of complex, divergent behaviour[53–55].

Key to our approach to build communicating artificial cells is the amplification of the signalling molecule in the Receivers. By designing such a signal amplification step into the target cells, we have engineered a communication system capable of efficient signal transduction over large distances. This uniquely addresses the critical problem of signal dilution in the extracellular environment,

and adopts a biomimetic approach to remedy this[56]. To advance such amplification strategies, the development of selective transporters and receptors that can be readily incorporated into synthetic compartments would offer significant advantages, as it enables the transduction of signalling over the membrane[57–60].

To summarize, we have developed a biomimetic communication pathway between populations of artificial cells that can efficiently signal over large distances by virtue of signal amplification. Sender cells converted ATP into AMP, which allosterically activated an enzymatic cascade in the Receiver cells to produce NADH. As NADH is an important energy-rich metabolite involved in a plethora of anabolic reactions, this

platform can be extended to generate various functional responses within the Receivers. Thus, the controlled construction of such artificial cell communities and further integration of this enzyme-based signalling pathway with DNA nanotechnology would potentiate the bottom-up research into the hierarchical complexity of biological communication networks.

## Methods

**Materials**. All chemicals were used as received unless stated otherwise. All lipids were obtained from Avanti Polar Lipids, except cholesterol, which came from Sigma-Aldrich. Glycogen phosphorylase b (EC 2.4.1.1) from rabbit muscle, apyrase (EC 3.6.1.5) from *Solanum tuberosum* (the isoform with ATPase/ADPase ratio of ~1) and αHL from *Staphylococcus aureus* were obtained from Sigma-Aldrich, α-phosphoglucomutase (EC 5.4.2.2) from Megazyme, and glucose-6-phosphate dehydrogenase (EC 1.1.1.49) from *Leuconostoc mesenteroides* from Worthington. Paraffin oil (liquid, 0.86 g cm$^{-3}$ at 20 °C.) was from JT Baker. All other reagents and solvents were of high-quality grade, purchased from commercial suppliers and used without further purification unless stated otherwise. Ultrapure water (Milli-Q) was obtained from a Merck Millipore Q-Pod system (≥18.2 MΩ) with a 0.22 μm Millipore Express 40 filter. HBS buffer contained 10 mM HEPES, 135 mM NaCl, 5 mM KCl at pH 7.4.

**General experimental conditions**. All experiments were performed at room temperature, unless indicated otherwise. All enzyme solutions were kept on ice during the procedures, whenever possible. Enzymes were purified and buffer exchanged twice using Amicon Ultra-0.5 ml 10 K MWCO spin filters, snap-frozen and stored at –80 °C until further use. Enzyme concentrations were based on the LOT-specific activity as indicated by the manufacturer with 1 U being equal to the formation of 1 μmole of product per min at the specified temperature and pH.

**Assembly of GUVs**. GUVs were prepared according to the droplet transfer method[41]. First, lipid stock solutions in paraffin were prepared by mixing lipid chloroform stock solutions and paraffin oil, heating this to 80 °C for 30–60 min and leaving it under vacuum overnight. These paraffin stock solutions were stored at –20 °C and used within two weeks. Paraffin stock solutions were mixed to obtain 200 μl of a solution with DOPC, POPC and cholesterol in a molar ratio of 35/35/30. 1% DSPE-PEG2000-biotin and 0.06–0.24% of DOPE-Lissamine Rhodamine B, DOPE-Cy5 or DOPE-carboxyfluorescein were incorporated for improved yields and barcoding of GUV populations, respectively. These solutions were sonicated at room temperature for 10 min in a bath sonicator and cooled on ice for >15 min. Next, 20 μl of inner phase solution (vide infra) was emulsified in 200 μl of lipid solution by strong vortexing for 25 s while turning the reaction tube to prevent sedimentation of the water droplet. Directly after, the emulsions were incubated on ice for 10 min. Subsequently, they were layered on top of 150 μl pre-cooled outer phase solution in a 1.5 ml plastic reaction tube and immediately centrifuged at 4 °C for 20 min at 3300 × g. GUVs were harvested by puncturing the tube at the position of the GUV pellet and collecting the aqueous layer. At this point, they were either loaded into the observation chamber or washed to ensure the removal of any unencapsulated material.

**Washing of GUVs**. Any unencapsulated material was removed by centrifugation at 1500 × g for 2 min (at 4 °C) and replacement of the supernatant by 40 ul of fresh outer phase solution, which was repeated once. Finally, the GUVs were carefully resuspended in fresh outer phase solution and transferred to the observation chamber.

**Inner phase and outer phase compositions**. The Receiver GUVs were loaded with the Response Cascade, which typically consisted of 1× HEPES-buffered saline (HBS), 0.20 M sucrose, 2.3 mM MgCl$_2$, 5.8 mM CaCl$_2$, 8 μM glucose-1,6-bisphosphate (GBP), 2.0 mM NAD$^+$, 2.0 mg ml$^{-1}$ glycogen, 0.5 U ml$^{-1}$ GPb, 5.0 U ml$^{-1}$ PGM, 5.0 U ml$^{-1}$ G6PDH. The Sender GUVs contained 1× HBS, 0.20 M sucrose, 2.3 mM MgCl$_2$, 5.8 mM CaCl$_2$, 8 μM GBP, 2.0 mM NAD$^+$ and 39-77 U ml$^{-1}$ apyrase. The outer phases had the same composition as the inner phases, but contained 0.20 M sorbitol instead of sucrose and did not include any enzymes nor glycogen. For detailed compositions of the inner and outer phases used to prepare the GUVs throughout this study, please refer to the Supplementary Methods.

**Monitoring NADH output in bulk reactions**. The Response Cascade was mixed with 0–5.0 mM Na$_2$HPO$_4$ and either 0–1.0 mM AMP or 15 U ml$^{-1}$ apyrase and 2.6 mM ATP or ADP. Reactions were monitored in a black 384-wells plate with transparent bottom (sample volume 30 μl) using a multiplate reader that recorded the NADH absorbance at 340 nm. NADH concentrations were calculated from a standard curve (Supplementary Fig. 21). The mean values and standard deviations of three samples are reported.

For the kymograms of the Response Cascade in absence of GUVs (Fig. 2h, i), the channel device containing the Response Cascade was imaged over its entire length using an objective with 10× magnification. To construct the kymograms, the images were lined up along the direction of AMP/phosphate diffusion and each image was further divided into four equal rectangular sections. The fluorescence

intensities of each section and each time point were extracted using Fiji and used to construct the kymograms.

**Channel device fabrication and coating**. A 1 mm wide channel was cut into three layers of adhesive Greiner Bio-One Easyseal. A 13 mm round coverslip was glued on top of this mold using double-sided tape. The channel mold was glued onto a 50 mm diameter #1.5 glass coverslip, taking care not to introduce air bubbles (Supplementary Fig. 22). Generally, three channel devices are glued on the same 50 mm coverslip for parallel imaging of multiple conditions. Channels were incubated with a 1 mg ml$^{-1}$ BSA solution in Milli-Q to passivate the surfaces and prevent lipid adsorption. To remove the BSA, the channels were washed 4× with outer phase solution. Whenever other microscope slides were used for imaging, they were also passivated with BSA and subsequently washed with Milli-Q.

**Imaging of GUVs**. The GUVs were loaded into BSA-passivated observation chambers: either an 8-well μslide with a #1.5 glass coverslip bottom (Ibidi), a 384 well Sensoplate Plus microplate with a #1.5 glass bottom (Greiner), or the custom-made channel device depicted in Supplementary Fig. 22. After sedimentation of the GUVs, αHL was added and incubated for up to 30 min. Next, Na$_2$HPO$_4$ and either AMP, ADP or ATP were added. Imaging was started as soon as the GUVs had settled again and performed every 20–40 min. For the channel device, to prevent rapid evaporation the GUVs were premixed with αHL and Na$_2$HPO$_4$ and quickly loaded into the channel. One side of the channel was sealed with two-component epoxy glue and AMP or ATP was added to the other side with minimal disturbance of the buffer in the channel. Finally, this side was sealed as well and images along the entire length of the channel were collected as soon as all GUVs had settled into position.

**Confocal laser scanning microscopy**. Imaging was performed on a Leica TCS SP5X inverted microscope equipped with HCX PL APO CS 10× NA 0.4 dry, 20× NA 0.7 dry and 40× NA 1.10 water-immersion objectives. All fluorophores were imaged using sequential scanning. NADH was excited using a Chameleon 2-photon laser (Coherent): $\lambda_{ex}$ = 725 nm, $\lambda_{em}$ = 395–475 nm. All other fluorophores were excited using a Leica white light laser: $\lambda_{ex}$ = 495 nm, $\lambda_{em}$ = 505–550 nm (DOPE-CF); $\lambda_{ex}$ = 555 nm, $\lambda_{em}$ = 565–630 nm (DOPE-LRB); $\lambda_{ex}$ = 631 nm, $\lambda_{em}$ = 650–750 nm (DOPE-Cy5). The image resolution was 1024 × 1024 pixels and scanning speed was 400 Hz. The pinhole was set to 1 Airy unit, except for the NADH channel where it was set to maximum to allow multiphoton imaging. Typically, 2–4× line averaging was applied.

**Image analysis**. Images were analyzed using Fiji to follow the activation of the Response Cascade[61]. NADH fluorescence intensities were extracted from the micrographs. For the bulk reactions, images were divided into four sections and the intensities were extracted and plotted vs. time and position. For the GUVs, their internal NADH production was either tracked manually while correcting for drift (Figs. 3a, 4a, 6d), or their positions were determined using the built-in particle analysis tool with subsequent extraction of the NADH production per GUV. For this, a threshold was applied to the images to eliminate noise, followed by smoothening and conversion to a binary mask. Overlapping GUVs were separated using the watershed algorithm and all particles of 10–100 μm were selected automatically. The obtained selection was manually checked to remove any areas that contained aggregates instead of GUVs. NADH fluorescence intensities inside the GUVs were extracted using the obtained selection and plotted vs. time and position (Fig. 3b–e)

For the experiment in Fig. 5, the images were acquired with >20% overlap to allow manual stitching of the images to reconstruct the whole channel. The NADH production was quantified from the NADH fluorescence intensity throughout the channel at various time points. For the image analysis, the channel was divided into nine sections of equal size, and the fluorescent signal was normalized.

**Reporting summary**. Further information on research design is available in the Nature Research Reporting Summary linked to this article.

## Data availability

The data that support the findings of this study are available from the corresponding author upon reasonable request.

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

## Acknowledgements

Dr. Liesbeth Pierson and dr. ir. Mark van Turnhout are thanked for their advice regarding microscopy. Prof. dr. ir. Tom de Greef is acknowledged for his critical reading of the manuscript. The Dutch Ministry of Education, Culture and Science (Gravitation program 024.001.035) and the ERC Advanced grant Artisym 694120 are acknowledged for funding.

## Author contributions

B.C.B. and J.C.M.v.H. conceived the project and designed the experiments. B.C.B. and J.E. performed the experiments. B.C.B. performed the data analysis. J.C.M.v.H.

supervised the project. B.C.B. and J.C.M.v.H. wrote the manuscript. All authors reviewed the manuscript.

## Competing interests

The authors declare no competing interests.
