## [Peer Review File · Nature Communications]

Reviewers' Comments:

Reviewer #1:

Remarks to the Author:

In this revised version of an earlier manuscript, the authors have responded appropriately to the reviewer comments (in particular to those of this reviewer), and improved the manuscript accordingly. It is now recommended to accept it for publication.

Reviewer #2:

Remarks to the Author:

The Authors replied to all my questions and addressed all my concerns.

Reviewer #4:

Remarks to the Author:

The manuscript presented by Dr. van Hest and collaborators focuses on letting simplified cell models (artificial cells) exchange chemical signals. The study is within the current synthetic biology research, more specifically, in the bottom-up branch.

The work is technically sound and the Authors provide strong evidence for their conclusions. Results are novel (in the sense that no one has reported this exact system before), but clearly the idea of chemical signalling between artificial cells (or between artificial cells and biological cells) is not new. The manuscript is interesting and valuable for specialists in the field of bottom-up synthetic biology.

Although the study does not deliver very strong messages to 'influence the thinking' in the field, I believe that this work will impact positively on the community and bring the artificial cell technology one step further. The design here presented by the authors is interesting, in my opinion, especially because it introduces the concept of signal amplification (via allosteric activation) which was missing in previous studies - generally based on simpler biochemical reactions, or on transcription factors activation. The approaches complement each other.

Looking at the previous Reviewer considerations, and at the Authors replies, the Authors have addressed all specific/technical comments.

My personal opinion is that this study, also thanks to the additional experiments carried out by the authors (for addressing Reviewer 1 criticism) is suitable for publication in Nature Communications, after major and minor revisions as suggested below:

Major revisions

M1) When the Authors start to list a series of papers about communicating artificial cells, I see - if I am not wrong - that the study by S. Mann is missing [ACS Synth Biol. 2018, 7(2):339-346]

M2) Line 88- and following. The Authors close the Introduction by recalling that the system under study is an information-processing system based on communication. My suggestion is to mention here that the use of artificial cells to develop new technologies based on information and communication was explicitly put forward several years ago, before the experimental papers commented by the authors, and specifically in [BioSystems 2012, 109, 24-34].

M3) Line 206, the interesting comment about the observed heterogeneity of enzyme-containing GVs made by the droplet transfer method recalls an important warning to give the readers: i.e., about the stochastic factors affecting solute content in that method (as well as in others) and

about the relevance of individual behavior within populations. I believe that this aspect should be stressed more in the current manuscript, carrying out population analyses with the available data (i.e., showing some intriguing element deriving from that heterogeneity). This will also add to the relevance of the current study (see comments by previous Reviewers). I do not mean here to perform extensive data analysis, but just to highlight, in quantitative manner and with key analyses, the important consequences of between-artificial cell heterogeneity (e.g., size dependence, distance dependence, or similar). Similarly, Figure S8 is also interesting in this respect. And the comment on line 434 of the SI-file. Thus, Authors definitely observed this relevant fact. A suggested reference that could be inserted to favor the readers is the following extensive review: [Synthetic Biology 2018, 3, ysy011]. This consideration is also relevant to counteract, more convincingly, to the negative evaluation of Reviewers 2 and 3 of the previous submission.

Minor revisions

m1) it would be interesting to see or to know from other sources the time-dependency of product distribution of apyrase (how much ADP, how much AMP)

m2) line 370, probably it is 'glucose' instead of 'sucrose'

m3) please explain the function of glucose-1,6-bisphosphate. Moreover, please be sure that all acronyms are defined at least in the SI-file

m4) Figure S1 is quite unclear. My suggestion is to add a picture of the device

m5) Caption of Figure S2: experiments in triplicate are mentioned, but it is not clear whether the shown points are from 3 different experiments or whether every point is the average of three experiments. In the latter case, error bars are missing.

m6) Figure 3d: the position of i-ii-iii-iv should be better specified in terms of mm (avoid "near" "far").

m7) line 356: please specify what kind of paraffin was used.

Reviewers comment

Our response

Reviewer #1:

In this revised version of an earlier manuscript, the authors have responded appropriately to the reviewer comments (in particular to those of this reviewer), and improved the manuscript accordingly. It is now recommended to accept it for publication.

We are grateful for the approval of our previous responses and improvements to the manuscript.

Reviewer #2:

The Authors replied to all my questions and addressed all my concerns.

We are content to have addressed your concerns appropriately.

Reviewer #4:

The manuscript presented by Dr. van Hest and collaborators focuses on letting simplified cell models (artificial cells) exchange chemical signals. The study is within the current synthetic biology research, more specifically, in the bottom-up branch.

The work is technically sound and the Authors provide strong evidence for their conclusions. Results are novel (in the sense that no one has reported this exact system before), but clearly the idea of chemical signalling between artificial cells (or between artificial cells and biological cells) is not new. The manuscript is interesting and valuable for specialists in the field of bottom-up synthetic biology.

Although the study does not deliver very strong messages to 'influence the thinking' in the field, I believe that this work will impact positively on the community and bring the artificial cell technology one step further. The design here presented by the authors is interesting, in my opinion, especially because it introduces the concept of signal amplification (via allosteric activation) which was missing in previous studies - generally based on simpler biochemical reactions, or on transcription factors activation. The approaches complement each other.

Looking at the previous Reviewer considerations, and at the Authors replies, the Authors have addressed all specific/technical comments.

My personal opinion is that this study, also thanks to the additional experiments carried out by the authors (for addressing Reviewer 1 criticism) is suitable for publication in Nature Communications, after major and minor revisions as suggested below:

Major revisions

M1) When the Authors start to list a series of papers about communicating artificial cells, I see - if I am not wrong - that the study by S. Mann is missing [ACS Synth Biol. 2018, 7(2):339-346]

We would like to thank the reviewer for their critical insight and for pointing out this omission. We have added a reference to the mentioned study in the introduction.

M2) Line 88- and following. The Authors close the Introduction by recalling that the system under study is an information-processing system based on communication. My suggestion is to mention here that the use of artificial cells to develop new technologies based on information and communication was explicitly put forward several years ago, before the experimental papers commented by the authors, and specifically in [BioSystems 2012, 109, 24-34].

Thank you for your suggestion. This relevant conceptual study has now been incorporated into the introduction.

M3) Line 206, the interesting comment about the observed heterogeneity of enzyme-containing GVs made by the droplet transfer method recalls an important warning to give the readers: i.e., about the stochastic factors affecting solute content in that method (as well as in others) and about the relevance of individual behavior within populations. I believe that this aspect should be stressed more in the current manuscript, carrying out population analyses with the available data (i.e., showing some intriguing element deriving from that heterogeneity). This will also add to the relevance of the current study (see comments by previous Reviewers). I do not mean here to perform extensive data analysis, but just to highlight, in quantitative manner and with key analyses, the important consequences of between-artificial cell heterogeneity (e.g., size dependence, distance dependence, or similar). Similarly, Figure S8 is also interesting in this respect. And the comment on line 434 of the SI-file. Thus, Authors definitely observed this relevant fact. A suggested reference that could be inserted to favor the readers is the following extensive review: [Synthetic Biology 2018, 3, ysy011]. This consideration is also relevant to counteract, more convincingly, to the negative evaluation of Reviewers 2 and 3 of the previous submission.

The reviewer raises an interesting point here, as many studies on Giant Vesicles as cell-mimetic compartments that incorporate an enzymatic reaction have reported a higher-than expected between-compartment variability of reaction rates, plateaus, and lag times. Indeed, we observe a relatively large heterogeneity in vesicle response rates as well.

While a full investigation of the underlying mechanisms falls outside the scope of this manuscript, we have performed further analyses on the kinetics of the response of individual vesicles, and accordingly can speculate on the most relevant factors contributing to these observations.

We have extended our discussion of the heterogeneity in the response of individual vesicles and included relevant references in the main manuscript. Additionally, we have added a Supplementary Note and further analyses of vesicle size vs. response to the SI (Suppl. Figs. 8, 13, 15).

Indeed, we did not observe a strong correlation with the size of the vesicles (Suppl. Fig. 8, 13). This is in line with the relatively large size (50 μm) and high enzyme concentrations (~ 100 nM) of the vesicles, which limit the expected contribution of external stochastic effects caused by solute partitioning during vesicle formation. However, as is clear from the excellent reviewing of the current literature in [Synthetic Biology 2018, 3, ysy011], the

variability in the response of enzymatic reactions inside Giant Vesicles produced by the droplet transfer method cannot be fully explained on the basis of stochastic solute partitioning only. Further disentanglement of the underlying mechanisms – such as confinement effects, protein–lipid interactions and the method of vesicle formation – is possible only by careful quantification of all relevant solute concentrations and interactions, as well as elucidation of the detailed mechanisms of vesicle formation. While extensive quantification of this complete set of parameters falls outside the scope of this manuscript, we do show that the vesicles have differential permeability constants (Suppl Fig. 15), which is attributed to the expected between-vesicle variability in the spontaneous formation of alpha-haemolysin pores.

Taken together, the observed between-vesicle variability in the response to AMP can most likely be attributed to larger-than expected (based on stochastic solute partitioning) variability in the GUV solute entrapment, as well as differences in the aHL content of individual vesicles due to spontaneous insertion and formation of membrane pores. While we do appreciate the highly interesting implications of between-compartment variability on the behavior of individual vesicles, the observed variability does not affect our overall conclusions based on the averaged responses of the entire population of vesicles.

Minor revisions:

m1) it would be interesting to see or to know from other sources the time-dependency of product distribution of apyrase (how much ADP, how much AMP)

The enzyme from *Solanum tuberosum* has two isozymes: one with an ATPase/ADPase ratio of ~10 and another with no selectivity (i.e. ATPase/ADPase activity = 1). Since the ADP is a non-active intermediate whose build-up would be undesirable, we opted for the isozyme with an ATPase/ADPase ratio of ~1, which liberates two equivalents of orthophosphate from ATP at identical rates. (*Arch. Biochem. Biophys.*, 93 (2), 353-363 and *Phytochemistry*, 21 (3), 551–558.)

This information is included in the SI.

m2) line 370, probably it is 'glucose' instead of 'sucrose'

We did mean ‘sucrose’ here, because the inner phases contain sucrose, whereas the outer phase does not: instead it contains sorbitol. Commonly glucose is used for the outer phase, but that was not possible here because of its inhibitory effect on the enzymes of the Response Cascade (see Line 191-194.)

m3) please explain the function of glucose-1,6-bisphosphate. Moreover, please be sure that all acronyms are defined at least in the SI-file

Glucose-1,6-bisphosphate is a commonly used activator of phosphoglucomutase that serves as an initial phosphate donor to phosphorylate the serine residue in the enzyme’s active site (see e.g. Lee et al. *J Biol Chem.* 2014 Nov 14; 289(46): 32010–32019).

We have added this information to the SI, as well as a full list of all acronyms used throughout the manuscript.

m4) Figure S1 is quite unclear. My suggestion is to add a picture of the device

We have updated the figure according to your suggestion: both a picture of the device and a cartoon showing the device from the side are now included for clarification.

m5) Caption of Figure S2: experiments in triplicate are mentioned, but it is not clear whether the shown points are from 3 different experiments or whether every point is the average of three experiments. In the latter case, error bars are missing.

We have added the appropriate error bars, representing the standard deviation of three independent samples.

m6) Figure 3d: the position of i-ii-iii-iv should be better specified in terms of mm (avoid "near" "far").

We have added this information to the figure's caption, in correspondence with the distances indicated in the scale bars of figure 3b,c.

m7) line 356: please specify what kind of paraffin was used.

We have added this information to the manuscript and SI.